# Impact of Hypoxia–Hyperoxia Exposures on Cardiometabolic Risk Factors and TMAO Levels in Patients with Metabolic Syndrome

**DOI:** 10.3390/ijms241914498

**Published:** 2023-09-24

**Authors:** Afina Bestavashvili, Oleg Glazachev, Shabnam Ibragimova, Alexander Suvorov, Alexandros Bestavasvili, Shevket Ibraimov, Xinliang Zhang, Yong Zhang, Chavdar Pavlov, Elena Syrkina, Abram Syrkin, Philipp Kopylov

**Affiliations:** 1World-Class Research Center “Digital Biodesign and Personalized Healthcare”, I. M. Sechenov First Moscow State Medical University, 119991 Moscow, Russia; suvorov_a_yu_1@staff.sechenov.ru (A.S.); kopylov_f_yu@staff.sechenov.ru (P.K.); 2Department of Normal Physiology, N.V. Sklifosovsky Institute of Clinical Medicine, I. M. Sechenov First Moscow State Medical University, 119991 Moscow, Russia; glazachev_o_s@staff.sechenov.ru (O.G.); zxl0620@yandex.ru (X.Z.); 3Department of Therapy of the Institute of Professional Education, I. M. Sechenov First Moscow State Medical University, 119991 Moscow, Russia; shabnam.olimp@mail.ru (S.I.); pavlov_ch_s@staff.sechenov.ru (C.P.); 4Department of Medicine, Salem Hospital, Mass General Brigham, Salem, MA 01970, USA; abestavasvili@mgb.org; 5Department of Cardiology, Functional and Ultrasound Diagnostics, N.V. Sklifosovsky Institute of Clinical Medicine, I. M. Sechenov First Moscow State Medical University, 119991 Moscow, Russia; sheva7864@gmail.com (S.I.); syrkina_e_a@staff.sechenov.ru (E.S.); previntenscardiology@staff.sechenov.ru (A.S.); 6The State-Province Key Laboratories of Biomedicine-Pharmaceutics of China, Key Laboratory of Cardiovascular Research, Ministry of Education, Department of Pharmacology, Harbin Medical University, Harbin 150081, China; hmzhangyong@hotmail.com; 7Department of Gastroenterology, Botkin Hospital, 125284 Moscow, Russia

**Keywords:** gut microbiome, intermittent hypoxic training, inflammatory response, lipid oxidation, metabolic syndrome, reactive oxygen species, trimethylamine-N-oxide

## Abstract

Along with the known risk factors of cardiovascular diseases (CVDs) constituting metabolic syndrome (MS), the gut microbiome and some of its metabolites, in particular trimethylamine-N-oxide (TMAO), are actively discussed. A prolonged stay under natural hypoxic conditions significantly and multi-directionally changes the ratio of gut microbiome strains and their metabolites in feces and blood, which is the basis for using hypoxia preconditioning for targeted effects on potential risk factors of CVD. A prospective randomized study included 65 patients (32 females) with MS and optimal medical therapy. Thirty-three patients underwent a course of 15 intermittent hypoxic–hyperoxic exposures (IHHE group). The other 32 patients underwent sham procedures (placebo group). Before and after the IHHE course, patients underwent liver elastometry, biochemical blood tests, and blood and fecal sampling for TMAO analysis (tandem mass spectrometry). No significant dynamics of TMAO were detected in both the IHHE and sham groups. In the subgroup of IHHE patients with baseline TMAO values above the reference (TMAO ≥ 5 μmol/l), there was a significant reduction in TMAO plasma levels. But the degree of reduction in total cholesterol (TCh), low-density lipoprotein (LDL), and regression of liver steatosis index was more pronounced in patients with initially normal TMAO values. Despite significant interindividual variations, in the subgroup of IHHE patients with MS and high baseline TMAO values, there were more significant reductions in cardiometabolic and hepatic indicators of MS than in controls. More research is needed to objectify the prognostic role of TMAO and the possibilities of its correction using hypoxia adaptation techniques.

## 1. Introduction

The effect of the gut microbiome, which has been described as a separate organ in some studies, on cardiometabolic pathology (atherosclerosis, hypertension, obesity, insulin resistance (IR)) is now well established [1,2,3,4,5]. A number of studies have shown that certain metabolites of the gut microbiome, such as increased trimethylamine-N-oxide (TMAO), which is formed in the liver after the splitting of food with a high content of choline, phosphatidylcholine, L-carnitine, and other trimethylamine (TMA)-containing nutrients, are associated with a high risk of the initiation and progression of coronary heart disease (CHD), heart failure, hypertension, type 2 diabetes mellitus (T2DM), obesity, and peripheral artery disease [6,7,8,9].

In particular, elevated plasma levels of TMAO have been shown to be associated with diastolic dysfunction, contributing to the development of early vascular atherosclerosis through accelerating endothelial injury and dysfunction [10,11]. TMAO initiates pro-atherosclerotic mechanisms: inhibition of reverse cholesterol transport by macrophages, increased platelet responsiveness and risk of thrombosis, and induction of inflammatory vascular endothelial damage [12,13,14,15]. Thus, several studies have shown a direct relationship between TMAO levels, inflammatory processes, atherosclerosis, T2DM, and other cardiovascular diseases (CVD) [12,13,16,17,18]. A systematic review and meta-analysis by Schiattarella et al. demonstrated that there is a positive dose-dependent correlation between blood TMAO levels and increased cardiovascular risk as well as overall mortality in humans [17,19]. According to Kanitsoraphan et al., high circulating blood TMAO levels were associated with a 2.07–2.7-fold higher overall mortality in patients with T2DM, including after adjustment for body mass index (BMI) [20].

Recent data demonstrate the potential role of the gut microbiota as a pathogenic factor associated with metabolic syndrome (MS). Qualitative and quantitative changes in the gut microbiota are associated with various clinical conditions, such as obesity, insulin resistance and T2DM, and steatohepatitis [21,22,23]. Barrea L. et al. have shown that circulating blood levels of TMAO correlate positively with the degree of obesity, as well as with other MS components, including a surrogate index of IR-HoMA-IR. When the markers of cardiometabolic risks and early predictors of nonalcoholic fatty liver disease (NAFLD), the Visceral Adiposity Index (VAI) and the Fatty Liver Index (FLI), were analyzed, their association with the gut microflora metabolite TMAO was also found [24,25]. In addition, Chen et al., in a study conducted in China and including a sample of 1688 patients diagnosed with NAFLD, a hepatic manifestation of MS, found an association between the presence and severity of NAFLD and TMAO levels, which they positioned as an independent marker and possible risk factor for NAFLD [25,26].

Given the significant potential involvement of TMAO in the development of cardiovascular and metabolic diseases, there is an active search for TMAO inhibitors. No direct inhibitors of TMAO have yet been found, but the compounds 3,3-dimethyl-1-butanol (DMB) and resveratrol are known to inhibit TMAO formation and its negative effects. Through inhibiting the formation of TMA by gut microbes, DMB thereby reduces plasma TMAO levels and also attenuates the development of atherosclerosis caused by a diet high in choline [27]. In addition, DMB treatment significantly reduced plasma TMAO levels and prevented the development of cardiac dysfunction in mice fed with a “Western” diet [28]. Resveratrol isolated from red wine attenuates TMAO-induced atherosclerosis via regulating TMAO synthesis and bile acid metabolism through gut microbiota remodeling [29].

In addition, it was found that episodes of “acute” hypoxia as well as a prolonged stay under conditions of medium-high altitude significantly and multi-directionally change the ratio of intestinal microbiome strains and their metabolites in feces and blood [30]. At the same time, a stay under moderate hypoxia conditions is accompanied by the inhibition of certain gut strains most actively producing TMA, which is considered a definite basis for the application of procedures for adaptation to hypoxia for targeted influence on TMAO production and its associated potential risk factors, CVD and MS.

Intermittent normobaric hypoxic–hyperoxic exposures (IHHEs), as one of the variants of the technology of adaptation to periodic hypoxia in recent years, are actively used in the treatment and complex rehabilitation of patients with multiple pathologies and comorbidities [31]. For example, it was found that a 3-week course of IHHE in CHD patients leads to a significant decrease in blood pressure and resting heart rate, and an increase in exercise tolerance [32]. Another study has shown that adding an IHHE course to the physical rehabilitation program in geriatric patients with initial dementia is accompanied by improved cognitive function and physical mobility [33].

Interval hypoxic–hyperoxic training procedures are a new approach in which the patient’s breathing through an oronasal mask with a hypoxic gas mixture is alternated with episodes of breathing with a hyperoxic mixture (FiO_2_ 35%), which shortens the reoxygenation period and stimulates the redox regulation system. It is known that hypoxia, short-term or long-term, triggers an HIF-induced response with the expression of genes and proteins (EPO, NOS, VEGF, glycolytic enzymes, etc.) providing vasculogenesis, mitochondrial biogenesis, and an increase in the adaptive capabilities of cells and the whole organism, while episodes of moderate hyperoxia are accompanied by the activation of ROS-induced mechanisms activating the antioxidant defense systems of cells, preventing the development of potentially harmful oxidative stress, as well as upregulating HIF-dependent adaptive mechanisms due to other molecular cascades and ROS sensors [34,35].

Intermittent hypoxic conditioning and IHHE, as one of effective hypoxic conditioning protocols, have been used in rehabilitation courses for a wide number of different diseases, but no studies have shown any significant adverse side effects [36]. A review with one of our co-authors on the use of IHHE in cardiological patients also showed no adverse side effects [37]. Data regarding the effects of passive intermittent hypoxic exposures on the metabolism of lipids, carbohydrates, and oxidative status in patients remain rather contradictory [36,38,39]. We have not found any data regarding the possible effects of hypoxic conditioning sessions in different modes on the state of the intestinal microbiome and the level of individual indicators of the gut microflora metabolism.

The aim of this pilot study was to evaluate the possible effects of IHHE on the metabolic product of the intestinal microbiome (TMAO) and the relationship of its dynamics with cardiometabolic risk factors and components of MS.

## 2. Results

At baseline, the compared groups did not differ in TMAO and most of the controlled variables, except for systolic blood pressure (SBP), total cholesterol (TCh), and low-density lipoprotein (LDL), which were significantly higher in the IHHE group (Table 1). After three weeks of IHHE interventions, there was a decrease in the levels of TMAO in the IHHE group, but the values were not significantly different from the controls.

Despite not having found changes in the levels of TMAO in both groups after IHHE interventions, the degree of reduction in most of the MS and inflammation indicators (Z-value) in the IHHE group was significantly higher than in the controls. And the values of SBP, diastolic blood pressure (DBP), fibrosis degree, as well as N-terminal pro-brain natriuretic peptide (Nt-proBNP) became significantly lower compared to the control in the post-intervention examination.

The majority of patients had no TMAO detected in the stool. In a small number of cases (9 patients out of 65), the presence of TMAO in stool samples was probably related to the consumption of products containing large amounts of TMAO or its precursors.

Table 2 shows the dynamics of TMAO and the analyzed cardiometabolic indicators of MS in IHHE patients divided into two subgroups: subgroup 0, with normal baseline TMAO values (<5.0 μM/L), and subgroup 1, or patients with elevated TMAO values (>5 μM/L). In the pre–post dynamics, there was a significant reduction in TMAO in subgroup 1 and reductions in all analyzed indicators in both subgroups. At the same time, the degree of reduction in TCh, LDL, and regression of liver steatosis index (Pre–Post Δ) was more pronounced in patients in subgroup 0, with initially normal TMAO values.

## 3. Discussion

The results of the study show that a three-week course of hypoxic conditioning in the IHHE mode against the background of a balanced medical treatment leads to a more pronounced regression in the values of known components of MS, such as TCh, LDL, BP, and an improvement of the liver fibrotic index than in the placebo group, and can also potentially have beneficial effects on a new marker of cardiovascular health—TMAO in patients with metabolic syndrome.

TMAO, which belongs to the class of aminoxides, is known as a product of the metabolism of the intestinal microbiome. Many studies have established a link between the metabolism of certain microbes and the development of cardiovascular disease. As early as 1988, Saikku et al. noted higher antibody titers to C. pneumoniae in patients who suffered from myocardial infarction [40]. Subsequently, a large cohort study of patients with coronary heart disease identified several metabolites, including trimethylamine-N-oxide, that significantly increase cardiovascular risk [17]. 

The adverse effects of TMAO are due to several metabolic pathways. TMAO contributes to the early pathological process of atherosclerosis through accelerating endothelial activation and dysfunction, including reduced endothelial self-renewal and increased monocyte adhesion through activation of PKC/NF-κB/vascular cell adhesion molecule-1 (VCAM-1) [41]. The proatherogenic effect of TMAO is also promoted by its adverse effect on cholesterol transport. TMAO inhibits a key enzyme of bile acid synthesis, 7-α-hydroxylase, thus reducing the formation of bile acids from cholesterol [42]. Another point of application of TMAO is the enterocytes, where trimethylamine (TMA), affecting the expression of C1-like protein Newman-Pick, ATP-linked cassette transporters class G1, G5/G8, inhibits cholesterol transport into the intestinal lumen and helps to increase its concentration in the blood [43]. Thus, the level of TMAO has a significant effect on the progression of the atherosclerotic process and is in close interaction with the activity of the intestinal flora.

Dysbiosis, particularly through an unbalanced diet with a high-content carbohydrate intake, contributes to oxidative stress through increased expression of proinflammatory cytokines. This leads to the oxidation of low-density lipoproteins (LDLs), affecting the production of NO and endothelin-1 [11,44]. At the same time, inflammation and oxidative stress are significant in the pathogenesis of the “hepatic component” of MS-NAFLD [45].

One of the mechanisms through which an unbalanced gut microbiota is involved in the pathogenesis and progression of NAFLD is an increase in the permeability of the intestinal wall, development of ischemic necrosis of mesenteric epithelial cells and induction of liver fibrosis development [46,47,48]. When NAFLD progresses to non-alcoholic steatohepatitis (NASH), changes in the gut microbiome in the form of its overgrowth are also observed [49], which leads to the hyperproduction of reactive oxygen species and increased risk of liver cancer development [50].

Given the role of TMAO in the progression of cardiometabolic disorders, various ways to suppress the production of TMAO are being developed, including antibiotic therapy, but the evidence base for them has not been formed [40].

The fact of different qualitative and structural ratios of the intestinal microbiota under different oxygen regimes seems to be interesting [50]. Thus, an increase in Prevotella species during prolonged systemic or local hypoxia has been noted, which contributes to intestinal dysfunction and inflammation [51,52]. 

On the other hand, moderate hypoxic stimuli can activate various transcriptional pathways via hypoxia response element (HRE) through activation of HIF-1 providing the production of protective proteins with antioxidant activity, inducible NOs, glucose transporters, glycolytic enzymes, selected growth factors, etc. [30]. Interestingly, short-term exposure to elevated oxygen conditions, interval hyperoxia, also exerts positive effects through other redox-dependent transcriptional cascades, particularly through the Nrf2 pathway and the antioxidant-response element (ARE), triggering the synthesis of antioxidants, anti-inflammatory cytokines, cellular matrix remodeling proteins, mitochondrial biogenesis, etc., providing a state of hormesis [30,53]. Precisely such a regime of hypoxic conditioning—IHHE was applied in our study, in which the patient’s breathing with a hypoxic gas mixture was interrupted with short intervals through feeding a hyperoxic gas mixture [32].

Based on the results obtained in this work and the described known effects of hypoxic conditioning, it is shown that the use of an IHHE course has a systemic optimizing effect on various metabolic and molecular mechanisms involved in the pathogenesis of MS. Despite the absence of a significant decrease in TMAO values, it can be reasoned that the normalization of lipid metabolism with a decrease in TCh and LDL, a reduction in CRP values, and a reduction in steatosis and liver fibrosis indicators are also partially associated with the improvement of the intestinal barrier and hepato-biliary functioning. This position corresponds to the work of Serebrovskaya et al. [34,54], where it has been shown that the use of IHHE or a course of IHHT reduces insulin resistance, normalizes carbohydrate metabolism in patients with prediabetes, and reduces the level of circulating proinflammatory molecules in patients with Alzheimer’s disease. The work of Timon et al., who noted a positive effect of 24-week hypoxic exposure on the inflammatory marker CRP in elderly people suffering from systemic chronic inflammation, also confirms this finding [55]. 

The possible involvement of TMAO in the pathogenesis of MS and NAFLD is indirectly confirmed by our data regarding the dynamics of TMAO and cardiometabolic indicators in a 3-week course of IHHE in patients with baseline elevated values of this metabolite (>5 μM/L). In the course of IHHE, their TMAO values decreased significantly, but the degree of reduction in TCh, LDL, and liver steatosis indicators was significantly less pronounced than in the group of patients with baseline normal TMAO values, which may indicate more significant changes in lipid metabolism and hepatic dysfunction and the need for longer IHHE courses to achieve the necessary effects in this group of patients.

It is also important to note the detected effects of IHHE in cardiohemodynamic indicators—a significant decrease in SBP and DBP, resting heart rate, and a significant decrease in the predictor of heart failure, i.e., NT-proBNP, which confirms our previous results, as well as the well-founded conclusion in the systematic review of Behrend T. et al. (2022), that IHHE or continuous sessions of hypoxia at rest or in combination with exercise are generally effective to reduce blood pressure and positively influence vascular health due to the stimulation of vascular adaptions (e.g., increased vascularization and endothelium-dependent vasodilatation) as well as adaptations in the autonomic nervous system disturbed by TMAO’s pro-atherogenic and pro-inflammatory activities in patients with MS [31,38]. 

To understand the positive dynamics of indicators of hepatic steatosis and fibrosis, the fact that hypoxia conditioning is thought to benefit smooth muscles in blood vessels but also elsewhere, including the bladder and gastrointestinal tract, mediated by diverse pathways involving heme oxygenase-1, NO, HIFs, VEGF, ATP-sensitive K+ channels, prostaglandins, bradykinin, different protein kinases, and ROS [55], normalizing the drainage and detoxification functions of the liver [56] and the barrier properties of the intestinal wall. The last is important given the proven direct cardiotoxic effect of molecules of microbial origin (lipopolysaccharides, peptidoglycans, and TMAO), which, interacting with receptors of cardiomyocytes and microenvironmental cells, can cause the development of myocardial remodeling and dysfunction, provoking the progression of heart failure [57]. In our work, on the other hand, we have shown a decrease in postload and NT-proBNP in the IHHE course. Moreover, the hypoxia conditioning-related priming of energy homeostasis molecules, e.g., adenosine, AMPK; the activation of systemic anti-inflammatory signaling, e.g., related to reduced levels of C3, CRP, and MCP-1; and the augmentation of antioxidant molecules, e.g., superoxide dismutase and glutathione [55], are potential contributors to hypoxia conditioning-associated cardiovascular and metabolic benefits.

In addition, changes in TMAO levels may be associated not only with changes in microbiome status, but also with changes in hepatic activity, as demonstrated by our results showing decreased TMAO levels and improved hepatic indicators, which were published earlier in our article [31]. 

### Limitations and Prospects

The sample size in the study was small. A larger study with no different baseline MS parameters should be conducted to obtain more reliable results. This was a single-centered trial using a limited number of patients, which does not allow firm conclusions in regard to rare but important safety events and side effects of IHHE.

Secondly, the patient’s dietary regime as well as physical activity were not closely monitored, which may have partly influenced both the dynamics of TMAO and the effectiveness of hypoxic conditioning.

Thirdly, TMAO values above the reference were found at baseline in only a small proportion of patients, which may have influenced the results of the study.

Finally, the study was limited to a survey before–post 3-week interventions; longer monitoring might have revealed the timing of the persistence of the observed effects of IHHE.

## 4. Materials and Methods

This prospective, single-center, randomized controlled trial enrolled 65 patients (32 women) with MS. Subjects’ ages ranged from 29 to 74 years. Patients were randomly allocated into groups: the IHHE group and the control group (sham). Patients in the IHHE group received intermittent hypoxic–hyperoxic treatment for 3 weeks, 5 days a week. The control group was simulated with the same protocol, but instead of intermittent hypoxia–hyperoxia, room air (normoxia, 20.9% O_2_) was given for the same training period.

The main endpoints were the evaluation of changes in plasma TMAO levels, the determination and evaluation of changes in TMAO stool samples, and the correlation detection of plasma and stool TMAO levels with other MS parameters. All endpoints were evaluated after three weeks of IHHE training.

We selected a number of parameters as secondary endpoints and compared baseline values with those after 3 weeks of exposure: hemodynamic parameters (systolic (SBP) and diastolic (DBP) blood pressure, heart rate (HR)), liver tissue elasticity and fibrosis stage according to liver elastography, lipid spectrum (total cholesterol—TCh, high- and low-density lipoproteins—HDL and LDL, triglycerides), and inflammatory markers (high sensitive C-reactive protein—CRP-hs, N-terminal prohormone brain natriuretic peptide—Ntpro-BNP).

The lipid profile and AST/ALT were measured using a Siemens Advia 1800 Biochemical Analyzer (“Siemens Healthcare Diagnostics Inc.”, Newark, DE, USA) and dedicated test kits “Siemens Healthcare Diagnostics Inc.”, USA. High-sensitivity C-reactive protein (CRP-hs, milligram per liter, “Beckman Coulter,” Brea, CA, USA) was measured using Siemens Advia 1800 Biochemical Analyzer (“Siemens Healthcare Diagnostics Inc.”, Newark, DE, USA) by immunoturbidimetry using latex particles. The levels of brain natriuretic peptide N-terminal prohormone (NTproBNP, “Biomedica,” Wien, Austria) were measured using an enzyme-linked immunosorbent assay and a Biochrom Anthos 2020 Jencons Microplate Reader photometer ("Biohrom Ltd.", Cambridge, UK). Weight and height were measured with a Seca gmbh&co.kg scale (Hammer Steindamm 3-25 22089 Hamburg, Germany). 

The assessment of the functional liver status and liver fibrosis stage and steatosis severity was performed by a noninvasive Vibration-Controlled Transient Elastography (VCTETM) ultrasound method. The measurements were done using the FibroScan 502 Touch (Echosense, Paris, France) device [31]. 

### 4.1. Methods

#### 4.1.1. Study Design

Subjects were recruited from March 2019 to March 2020 from the Cardiology Clinic of the First Sechenov Moscow State Medical University, Moscow, Russia. Laboratory and instrumental procedures, as well as the training course, were conducted in the same clinic. 

The study was approved by the ethical committee of the I.M. Sechenov First Moscow State Medical University (Local Ethical Protocol No. 05-19 10.04.2019) and was performed according to the ethical standards based on the Declaration of Helsinki—Ethical Principles for Medical Research Involving Human Subjects. Written informed consent was obtained from all trial participants. The study was registered at ClinicalTrials.gov (NCT04791397, protocol identifier A0519).

#### 4.1.2. Participants and Randomization

Eighty-six patients with MS aged 29 to 74 years who had been in a stable clinical condition within the past 3 months were invited to participate in the study. MS was defined according to National Institutes of Health guidelines as the presence of three or more of the following characteristics: waist circumference greater than 102 cm in men and greater than 89 cm in women, blood pressure ≥ 130/85 mm Hg for SBP and DBP, dyslipidemia (triglyceride level ≥ 150 mg/dL (1.7 mmol/L), high-density lipoprotein cholesterol < 40 mg/dL (1.04 mmol/L) in men or <50 mg/dL (1.3 mmol/L) in women, and elevated fasting blood glucose levels (≥100 mg/dL [5.6 mmol/L]). All subjects had the following exclusion criteria: individual intolerance to hypoxia, cirrhosis, Child–Pugh class C, patients serologically positive for hepatitis B and C, chronic kidney disease (GFR < 30 mL/min/1.73 m^2^), pregnancy, serious respiratory distress, acute cardiovascular disease, and neuromuscular disorders.

The study included 65 patients who were randomly assigned to either the IHHE group (32 patients) or the sham group. Withdrawal of consent, intolerance to hypoxia, territorial inconvenience, and inability to perform liver ultrasonography were reasons for excluding 21 patients from the study. We performed a simple blinded prospective randomized controlled study in two parallel groups. Participants were randomized using a 1:1 allocation ratio. Baseline anthropometrics, clinical characteristics, and medications are presented in Table 3. 

Groups were matched for gender, age, presence of MS components, and comorbidities. Patients were asked to adhere to their daily food intake, physical activity, prescribed medications, and habitual lifestyle throughout the study period.

#### 4.1.3. Program of Interval Hypoxic–Hyperoxic Exposures (IHHE)

Patients included in the study underwent a course of interval hypoxic–hyperoxic therapy using a special ReOxy-Cardio device (AI Mediq S.A., Luxemburg, Luxemburg). At the first visit, after all anthropometric and laboratory-instrumental examinations, the patients of both groups underwent a hypoxic challenge test (HCT) for 10 min. The test was performed in a sitting position in a chair, with hypoxic gas fed through a facial mask with a minimum oxygen content of 11% under the control of arterial blood oxygen saturation (SpO_2_) and heart rate, which were measured continuously using a built-in KIT Masimo pulse oximeter (measurement error ± 2%). The device automatically adjusted the individual mode of IHHE procedures for each patient based on the result of the HCT. Data on each procedure, starting from the HCT, were collected on the device. Starting from the second day of the study, the patients in the IHHE group were trained with a gas mixture with interval changes in O_2_ level from 10–14% (corresponding to 4000–6500 m above sea level) to 35% and nitrogen (N_2_). The patients inhaled on average a mixture of hypoxic gases with 11–12% O_2_ content for 4–7 min (depending on individual parameters set during HCT), followed by subsequent exposure to a hyperoxic gas mixture with 30–35% O_2_ content for 2–4 min.

The physician-researcher was present at each training session and monitored HR and SpO_2_, which were recorded and transmitted to the device monitor. When the individual minimum set SpO_2_ level (83–85%) was reached, the device was automatically switched to a hyperoxic gas mixture until full pre-hypoxic SpO_2_ recovery occurred (during 1–3 min). After that, the next cycle of hypoxia–hyperoxia was repeated. The training duration ranged from 40 to 45 min in each group and had no differences for anyone other than the researcher who conducted the session. Blood pressure was measured before and at the end of each treatment, using an automatic tonometer AND UA-767 (AND, Tokyo, Japan). 

Patients in the control group were trained according to the same scheme as patients in the main group, with the same “exposure time” and a number of workouts, but with a normoxic gas mixture administered throughout the session (room air, with an O_2_ concentration of 20.9%).

The total number of training sessions for patients in both groups was 15 hypoxic–hyperoxic or sham treatments, which were performed five times a week with a weekend break (2 days) for three weeks.

At the beginning of treatment, some patients in the IHHE group had brief complaints of light dizziness, which did not require the interruption of the procedure. After the first session, no patient refused to continue participation in the study.

### 4.2. Procedures 

Before IHHE or sham treatment, all patients underwent a routine medical examination, with a collection of medical history, data on the intake of drugs, family history for cardiovascular diseases, carbohydrate metabolism disorders, diabetes mellitus, and other diseases. One day before the experimental sessions, and 2–3 days after the last training, the patients of both groups underwent the same examinations and measurements: an evaluation of hemodynamic parameters (blood pressure (SBP and DBP) at rest, heart rate (HR)), SpO_2_, anthropometric data (height, body weight, waist, and hip circumference), measurement of liver tissue elasticity (stiffness), and evaluation of fibrosis stage as described with details in our previous study [31]. 

TMAO was measured in plasma and fecal samples, as described below. In addition, blood samples were collected to determine serum lipid levels (TCh, HDL, LDL, Mmol/L); liver enzymes—alanine aminotransferase (ALT, units per liter) and aspartate aminotransferase (AST, units per liter); markers of chronic inflammation—CRP-hs (mg/L), heat shock proteins-70 (HSP70, ng/mL); and N-terminal prohormone of brain natriuretic peptide (NTpro-BNP, pmol/L), and were reported in our previous study [39]. Samples (10 mL of venous blood) were collected (into vacuum lithium heparin and EDTA tubes) in the morning between 7 and 10 a.m., after an overnight fast of at least 8 h. In order to minimize platelet count, blood was allowed to clot (BD Vacutainer Plus SST), and serum separated immediately (by centrifugation at 3500 rpm for 15 min after sampling, Eppendorf Cen- trifuge 5702R, Darmstadt Germany), aliquoted, and stored at −80 °C until being processed. All biochemical analyses were conducted in the University’s hospital Blood Analysis Center, a blood biochemistry laboratory certified by the Moscow Department of Health Care. 

### 4.3. Metabolite Profiling 

#### 4.3.1. TMAO Plasma Quantification 

Plasma aliquots for TMAO analysis were isolated from whole blood, collected in tubes with EDTA, and stored at −80 °C from March 2019 to March 2020. One study indicated that TMAO is stable under these storage conditions for several years [58].

#### 4.3.2. Sample Preparation

Samples were thawed on ice and were vortexed for 1 min at +4 °C. Then, samples were centrifuged for 5 min at 5000× *g* rpm at +4 °C. A 20 μL aliquot was taken from each of the samples, after which the samples were re-frozen. The aliquot was mixed with 20 μL of an ice-cold (−40 °C) methanol solution with an internal standard.

The prepared samples were vortex-mixed for 5 min at +4 °C. Afterward, the samples were centrifuged for 10 min at maximum rpm (18,000× *g* rpm) at +4 °C. We took 25 µL of supernatant from the samples, transferred them to vials with screw-on lids, sealed them, and transferred for analysis using high-performance liquid chromatography-mass spectrometry (HPLC-MS). 

A Sciex 4500QTRAP tandem mass spectrometer (AB Sciex Pte. Ltd., Framingham, MA, USA) with a Shimadzu Nexera 30AD chromatograph (Shimadzu, Kyoto, Japan) was used for analysis. Ion source settings were as follows: temperature = 550 °C; GS1 = 55; GS2 = 55; CUR = 25; IS = 5500. Ion detection was performed in the positive ionization mode of the sample. 

The chromatographic separation of the components of the test sample was performed via RPLC chromatography using an Agilent ZORBAX Eclipse Plus (Aglient Technologies, Santa Clara, CA, USA) C18 2.1 × 50 mm 1.8 μm chromatographic column: phase A (water; 5 mM ammonium formate); phase B (ACN: MeOH 1:1; 5 mM ammonium formate); sample input volume 1 µL. Chromatographic gradient: 0 min 2% B; 0.25 min 2% B; 1.2 min 98% B; 2.5 min 98% B; 2.6 min 2% B; 3.5 min 2% B; flow 0.45 mL/min, thermostat temperature 45 °C. 

The referent values for TMAO were presumed to be 0.5–5.0 μM/L as recommended in certain studies [59,60]. 

#### 4.3.3. ТМАО Fecal Quantification

Fecal material samples for TMAO analysis were stored at −80 °C from March 2019 to March 2020.

#### Sample Preparation

A representative sample weighing approximately 20 mg was taken from the starting material in the frozen state and transferred into a sterile Eppendorf pipette. Afterward, 200 µL of an aqueous solution of the internal standard in phosphate buffer was added to the sample. The resulting solution was placed in a vortex at +4 °C and centrifuged for 10 min at 1400× *g* rpm. The resulting homogenate was recovered following centrifugation at 4000× *g* rpm for 10 min to maintain the integrity of the microbiota cells. Then, 100 µL of the resulting supernatant was transferred to an Eppendorf pipette with the addition of 20 µL of methanol and 120 µL of chloroform. The resulting solution was vortexed for 10 min at 1400× *g* rpm at +4 °C, after which the sample was centrifuged at a maximum of 18,000× *g* rpm for 10 min. The top layer of liquid was collected, transferred to vials with inserts and screw-on lids, and given for analysis. A Sciex 4500QTRAP tandem mass spectrometer with (AB Sciex Pte. Ltd., Framingham, MA, USA) a Shimadzu Nexera 30AD chromatograph (Shimadzu, Kyoto, Japan) was used for analysis.

### 4.4. Statistical Methods

All data were analyzed using Python Software Foundation version 3.8 for Windows (DE, USA). The data were reported as median and inter-quartile range 25–75. The assumption of normality and homoscedasticity was verified using Shapiro–Wilk’s W-test before parametric tests. To identify the magnitude of the statistical difference between sham and IHHE treatments in each period of the study, Fisher’s exact test was used. Because of the non-normal distribution of data, we used Mann–Whitney’s U-test or Wilcoxon’s test to compare the baseline data between the groups as well as different changes (normalized delta pre–post, z-scores) between the groups. Pearson’s or Spearman’s correlation analysis was performed to test relationships between the variables. The a-level was set at 0.05 for all statistical analyses.

## 5. Conclusions

Despite significant interindividual variation and a small number of patients with MS and baseline elevated TMAO in the study, it was noted that a hypoxic conditioning course in the mode of intermittent hypoxic–hyperoxic exposures at rest leads to a downward trend in blood TMAO, which was accompanied by a more significant reduction in cardiometabolic and hepatic indicators of MS than in the placebo group. Patients with baseline elevated values of this metabolite in the IHHE course showed a significant reduction in TMAO but a smaller degree of reduction of total cholesterol, low-density lipoproteins, and hepatic steatosis than in the subgroup of patients with baseline normal TMAO values. 

IHHE can potentially be considered as an effective and safe (well-tolerated) technique to improve cardiometabolic health, presumably through normalization of the gut microbiome, the barrier function of the intestinal wall, and hepatic metabolism.

More research is needed to objectify the role of microbiome dysfunction and TMAO in the development and progression of metabolic syndrome components and hepatic and cardiovascular pathology, as well as to identify the cellular and molecular mechanisms of interval hypoxic exposures to TMAO levels as an indicator of the “microbiome-lipid metabolism-oxidative stress-cardiovascular risk factors” interplay. The aspect of individualization of IHHE regimens with regard to the optimal combination of procedure variables to achieve the best physiological and structural adaptations deserves special attention.

## Figures and Tables

**Table 1 ijms-24-14498-t001:** Pre- and post-intervention data for cardio-metabolic parameters with the main analysis of covariance results and normalized deltas.

Variables	Groups	Pre-Test	Post-Test	*p*-Value **	Pre–Post Δ, z-Value	*p* Value (Mann–Whitney U) ***
SBP, mm Hg	IHHE	148 [136; 164] * *p* = 0.024	131.5 [121.7;144.0] * *p* = 0.012	<0.001	−0.61 [−1.21; −0.45]	<0.001
	Sham	142.5 [127.5; 152.2]	141.0 [130.0; 151.2]	0.161	0.08 [−0.14; 0.32]	
DBP, mm Hg	IHHE	92.0 [85.7; 102.0]	84.5 [79.7; 93.2] * *p* = 0.043	<0.001	−0.48 [−0.87; −0.29]	<0.001
	Sham	88.0 [81.5; 96.2]	92.0 [83.0; 98.2]	0.463	0.14 [−0.02; 0.48]	
Heart Rate, bpm	IHHE	71.0 [61.0; 77.2]	64.0 [58.0; 70.2]	<0.001	−0.39 [−0.54; −0.07]	<0.001
	Sham	66.5 [61.0; 75.7]	68.5 [60.7; 77.2]	0.561	0.00 [−0.15; 0.22]	
Total Cholesterol, Mmol/L	IHHE	5.86 [5.14; 6.85] * *p* < 0.001	5.01 [4.31; 5.86]	<0.001	−0.41 [−0.84; −0.25]	<0.001
	Sham	4.62 [3.72; 5.69]	4.93 [4.40; 5.73]	0.722	0.03 [−0.00; 0.26]	
LDL-Cholesterol, Mmol/L	IHHE	3.83 [2.77; 4.57] * *p* < 0.001	2.86 [1.95; 3.82]	<0.01	−0.51 [−0.75; −0.29]	<0.001
	Sham	2.65 [1.75; 3.56]	2.98 [2.29; 3.74]	0.763	0.04 [0.00; 0.28]	
TMAO, μM/L	IHHE	3.25 [2.32; 5.16]	2.76 [2.07; 3.46]	0.011	−0.16 [−0.57; 0.03]	0.320
	Sham	3.09 [2.15; 3.67]	2.73 [2.54; 3.74]	0.724	−0.14 [−0.32; 0.22]	
Liver Steatosis, stage	IHHE	7.50 [5.05; 11.10]	5.20 [3.45; 6.67]	<0.001	−0.14 [−0.29; −0.05]	<0.001
	Sham	6.05 [4.07; 9.90]	6.45 [4.40; 9.45]	0.651	0.00 [0.00; 0.02]	
Liver Fibrosis (LSM), kPa	IHHE	1.00 [0.00; 2.25]	0.00 [0.00; 1.00], *p* * = 0.025	<0.001	−0.24 [−0.37; −0.08]	0.420
	Sham	0.50 [0.00; 2.00]	1.00 [0.00; 2.00]	0.701	0.15 [0.05; 0.45]	
CRP-hs, mg/L	IHHE	2.33 [1.50; 4.30]	1.97 [1.15; 2.69]	<0.001	−0.07 [−0.45; 0.00]	0.031
	Sham	1.88 [1.51; 3.96]	2.36 [1.62; 4.52]	0.633	0.00 [−0.05; 0.07]	
NTproBNP, pmol/L	IHHE	5.83 [3.00; 17.2]	3.33 [3.00; 13.4] * *p* = 0.012	0.124	−0.03 [−0.12; 0.00]	<0.001
	Sham	11.2 [3.09; 32.1]	15.9 [5.87; 40.6]	<0.01	0.05 [0.00; 0.19]	

Values are expressed as median and first–fourth quartiles. *—significant difference between the groups at the same study time (Mann–Whitney U); **—significantly different pre- vs. post-treatment; ***—significantly different in Z-values between the groups. SBP—systolic blood pressure; DBP—diastolic blood pressure; TMAO—Trimethylamin-N-oxide; LSM—liver stiffness measurement; LDL—cholesterol, low-density lipoprotein; CRP-hs—high-sensitivity C-reactive protein; NTproBNP—N-terminal prohormone of brain natriuretic peptide.

**Table 2 ijms-24-14498-t002:** Pre- and post-intervention data for cardio-metabolic parameters with the main analysis of covariance results and normalized deltas for IHHE subgroups (Subgroup 0—TMAO < 5 μM/L, n = 24; subgroup 1—TMAO > 5 μM/L, n = 8).

Variables	Groups	Pre-Test	Post-Test	*p*-Value **	Pre–Post Δ, z-Value	*p* Value (Mann–Whitney U) ***
SBP, mm Hg	Subgroup 0	150.0 [141.0; 161.0]	130.5 [121.5; 141.0]	<0.001	−0.631 [−1.441; −0.439]	0.317
	Subgroup 1	137.0 [132.7; 167.2]	133.0 [125.5; 148.0]	<0.001	−0.467 [−0.864; −0.261]	
DBP, mm Hg	Subgroup 0	93.0 [87.5; 102.0]	84.5 [79.7; 92.2]	<0.001	−0.601 [−1.155; −0.346]	0.074
	Subgroup 1	89.500 [83.0; 100.2]	85.0 [79.5; 95.7]	0.741	−0.277 [−0.531; −0.185]	
Heart Rate, bpm	Subgroup 0	71.5 [64.0; 78.2]	64.5 [58.0; 69.7]	<0.001	−0.350 [−0.594; −0.192]	0.043
	Subgroup 1	68.5 [60.2; 73.2]	62.5 [60.0; 70.2]	0.053	−0.070 [−0.227; 0.000]	
Total Cholesterol, Mmol/L	Subgroup 0	5.54 [4.82; 6.60]	5.32 [4.30; 5.86]	<0.001	−0.359 [−0.674; −0.190]	0.018
	Subgroup 1	6.06 [5.75; 7.57]	4.72 [4.44; 5.65]	<0.001	−0.934 [−1.479; −0.719]	
LDL-Cholesterol, Mmol/L	Subgroup 0	3.77 [2.71; 4.55]	3.16 [1.93; 3.82]	<0.001	−0.410 [−0.588; −0.263]	0.010
	Subgroup 1	3.83 [3.54; 5.41]	2.71 [2.49; 3.12]	0.017	−0.827 [−1.677; −0.714]	
TMAO, μM/L	Subgroup 0	2.70 [2.24; 3.38], *p* * < 0.001	2.72 [1.86; 3.28]	0.495	−0.040 [−0.171; 0.047]	<0.001
	Subgroup 1	10.25 [9.25; 11.54]	2.81 [2.48; 4.78]	0.008	−1.808 [−2.031; −1.163]	
Liver Steatosis (LSM), stage	Subgroup 0	9.15 [5.17; 14.12]	5.20 [3.80; 6.72]	<0.001	−0.235 [−0.386; −0.123]	0.007
	Subgroup 1	6.20 [4.85; 7.30]	5.00 [3.30; 6.30]	0.008	−0.087 [−0.136; −0.047]	

The study did not reveal any correlations of baseline blood TMAO levels and pre–post dynamics with the values of the main indicators of MS, parameters of chronic inflammation, and liver steatosis. A weak correlation was revealed only between levels of TMAO at baseline and acetyl-carnitine (r = 0.30, *p* = 0.01), which can be explained by their formation from a common carnitine precursor [15]. *—significant difference between the groups at the same study time (Mann–Whitney U); **—significantly different pre- vs. post-treatment; ***—significantly different in Z-values between the groups.

**Table 3 ijms-24-14498-t003:** Anthropometric and clinical characteristics of the patients.

Variables	IHHE Group (n = 32)	Sham Group (n = 33)	*p*-Value
Gender, male	14 (43.7%)	19 (57.5%)	NS
Age, years	56.9 ± 11.7	59.8 ± 10.3	NS
Smoking	10 (31.2%)	11 (33.3%)	NS
Hypertension	32 (100%)	33 (100%)	NS
Diabetes Mellitus, type 2	12 (37.5%)	10 (30.3%)	NS
Obesity (BMI > 30)	24 (75.0%)	27 (81.8%)	NS
Weight, kg	92.0 (81.0; 114)	92.5 (82.8; 104)	NS
BMI, kg|m^2^	34.3 (30.2; 38.0)	32.4 (30.8; 35.8)	NS
Waist Circumference, cm	114 (108; 124)	110 (105; 120)	NS
Hip Circumference, cm	113.5 (106.8; 119.5)	110 (105;120)	NS
**Regular Medication:**			
Aspirin	4 (12%)	8 (24.4%)	NS
ACE inhibitors	16 (50%)	13 (39%)	NS
AT II inhibitors	14 (43.7%)	16 (48.4%)	NS
Calcium channel blockers	12 (37.5%)	12 (36.3%)	NS
Beta-blokers	14 (43.7%)	13 (39.9%)	NS
Diuretics	14 (43.7%)	14 (42.2%)	NS
Statins	14 (43.7%)	17 (51.6%)	NS
Metformine	14 (43.7%)	10 (30%)	NS
Sulfonylureas	6 (18.7%)	4 (12.1%)	NS
Insulin	2 (6.25%)	5 (15%)	NS

Note. The data are expressed as mean ± SD or frequencies (%). IHHE—intermittent hypoxic–hyperoxic training. BMI—body mass index. NS—nonsignificant differences between the groups.

## Data Availability

Raw data supporting the conclusions of this publication will be made available by the authors without undue reservations.

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
