# Peer review of "Impact of Hypoxia–Hyperoxia Exposures on Cardiometabolic Risk Factors and TMAO Levels in Patients with Metabolic Syndrome"

_ijms, 2023, doi:10.3390/ijms241914498_

Round 1

Reviewer 1 Report

The manuscript is original research evaluating the relationship between gut microbiome metabolites, particularly tri-methylamine-N-oxide (TMAO), and their potential role in cardiovascular disease (CVD) within the context of hypoxia preconditioning. Given the relevance of this topic and the study's prospective randomized design, it holds promise for furthering our understanding of CVD risk factors and the potential therapeutic use of hypoxia adaptation techniques. Nonetheless, several aspects require clarification and elaboration to strengthen the manuscript's impact and validity before considering it for publication.

The authors must review the manuscript title to accurately represent the research focus and avoid potential misconceptions among readers. While the study sheds light on the crucial role of TMAO levels and their association with cardiometabolic risk factors, it is essential to acknowledge that TMAO is just one of the many metabolites influenced by the gut microbiome. Relying solely on TMAO as an indicator of the microbiome status might lead to an incomplete understanding of the complex interactions within the gut ecosystem. TMAO levels can indeed be affected by hepatic metabolism, particularly through the activity of flavin monooxygenases in the liver. Thus, variations in TMAO levels might not solely stem from alterations in the gut microbiota but also from differences in hepatic enzymatic activity. Consequently, attributing all changes in TMAO to the gut microbiome alone might overlook significant contributions from hepatic processes. Furthermore, the "microbiome status" encompasses a broader spectrum, including microbial diversity, abundance of specific bacterial taxa, and functional capacity within the gut ecosystem. Isolating TMAO as the sole representative of the microbiome status may not fully capture the multifaceted influence of the gut microbiota on cardiometabolic health. Discuss this further.

Minor Changes:

1. Remove citations from the abstract.

2. Replace numerical numbers at the beginning of sentences with their corresponding word form. For example, "33" should be written as "Thirty-three."

3. Avoid mentioning commercial brand names in the abstract.

4. Arrange the keywords in alphabetical order.

Reviewer 2 Report

Authors have attempted to examine whether IHHE can improve cardiometabolic health of patients with MS via the reduction of TMAO. However, the use of IHHE is not well justified in the paper. IHHE is only briefly mentioned in the second last paragraph, and authors have spent most part of the introduction talking about TMAO and its relationship with cardiometabolic health. The evidence of advocating IHHE has not been clearly presented despite a few trials being mentioned, which is not related to the scope of this study (e.g. improved cognitive function and physical mobility). Adverse events and side effects of IHHE from previous studies were not presented neither. In the main study, authors have failed to detect significant differences in many of the comparisons, and TMAO, one of the key parameters in this study, was absent in majority of the stool samples. The study finding is probably so preliminary for publication in a recognized international journal like IJMS.

Nil

Reviewer 3 Report

Bestavashvili and colleagues investigated the association between trimethylamine-N-oxide (TMAO) and hypoxia treatment in patients with metabolic syndrome. In a prospective, randomized study, 65 patients with MS were enrolled and divided into two groups (33 treated and 32 placebo). Before and after IHHE treatment, patients underwent liver elastometry, biochemical blood tests, and blood and stool sampling for TMAO analysis. No significant TMAO dynamics were detected in either the IHHE or sham groups. In the subgroup of IHHE patients with baseline TMAO levels above the reference value (TMAO≥5 μmol/l), plasma TMAO levels were significantly reduced. However, the regression of TCh, the rate of LDL reduction and the liver steatosis index were more pronounced in patients with initially normal TMAO levels. Despite the significant interindividual variation, the results of his study showed that the subgroup of IHHE patients with MS and high baseline TMAO values had a more significant reduction in cardiometabolic and liver function indicators than the control group.

Comments and suggestions:

-          The title is misleading as the microbiome status was not investigated. I suggest revising it.

-          Reference in the abstract should be deleted.

-          There are several abbreviations (even if known) in the manuscript that have not been introduced (SBP, TCh, etc). These should be explained when they are first used.

-          The tables are numbered incorrectly (starting with Table 2).

-          Why are all the components of metabolic syndrome (abdominal volume, fasting blood glucose, etc) not shown in Tables 2 and 3?

-          Stylistic error, the p-value is written in both lower case and upper case in the manuscript, and some places to 2 and in others to 3 decimal places. I suggest a standardized presentation.

-          The description in lines 326 and 327 is incorrect: "blood pressure ≥130/85 mm Hg for BP and BP". Systolic and diastolic blood pressure would be indicated I suppose.

-          The description of the 65 persons selected into two groups is not clear. The authors also describe random selection and matching by gender, age and MS component. Line: 335 – 344.

-          Why is the proportion of people with diabetes lower in the IHHE group? And could this bias the results?

-          Why were there no weekly measurements during the three-week study period? This would allow monitoring of the trend of changes.

-          A table comparing the basic characteristics of the two groups based on TMAO (<5uM/L vs >5uM/L) is missing. In its absence, the results in Table 3 cannot be considered.

-          The hypoxia treatment had a significant positive effect on the lipid profile as shown in Table 2 but showed no significant correlation with TMAO levels. Thus, it is also questionable whether there is a direct relationship between hypoxia and TMAO or only an indirect effect through the lipid profile.

-          The authors themselves write that the results presented in this manuscript are based on a pilot. What is the reason for not continuing the research in the three years since its completion (March 2020)?

Overall, the research presented in the manuscript has several serious limitations, which the authors also note in the limitation subsection. One of the most serious of these is the small sample size and the lack of adjustment for dietary habits. Due to the questionable nature of the results presented, I do not recommend the manuscript for publication.

Round 2

Reviewer 2 Report

Nil

Nil

Author Response

Dear editorial board of "MDPI IJMS",
Thank you for receiving our manuscript and considering it for review. 

We are thankful to reviewer 2 for his analysis of our work. We have made changes to the manuscript based on the comments and recommendations.  

Reviewer 3 Report

The authors have not provided fully acceptable answers to many of the questions and comments I have raised.

1.       In the case of the revised manuscript, it is not clear what changes have been made. Please indicate the changes by highlighting or using the track changes function.

2.       There is also a lack of consistency in the way p-values are indicated, and in many cases, authors use a comma rather than the English decimal point.

3.       The justification for not including the components of the metabolic syndrome, that this has already been published in another article, is not acceptable. Please indicate them in the relevant tables.

4.       I do not find the term "corrected", often used by authors, to be appropriate for all the questions and comments I have raised. Please respond to my points with sentences.

5.       Does the difference in the proportion of people with diabetes between the two groups bias the results? Please explain your answer in detail.

6.       I maintain that the small sample size is a major limitation of the present study. The authors plan to repeat their study with a larger sample size. What do you expect from analyses with larger sample sizes?

Author Response

Dear editorial board of «MDPI IJMS»,

Thank you for receiving our manuscript and considering it for review.

We are grateful to reviewer 3 for his detailed analysis of our work! His comments are useful for planning our further study and improving the present manuscript.

Regarding his comments, we would like to respond the following:

The authors have not provided fully acceptable answers to many of the questions and comments I have raised.

  1. In the case of the revised manuscript, it is not clear what changes have been made. Please indicate the changes by highlighting or using the track changes function.

Part of the changes made on the reviewer's recommendation are highlighted in the text of the manuscript and the rest of the changes were made using the change tracking function.

  1. There is also a lack of consistency in the way p-values are indicated, and in many cases, authors use a comma rather than the English decimal point.

Changes have been made to all tables.

  1. The justification for not including the components of the metabolic syndrome, that this has already been published in another article, is not acceptable. Please indicate them in the relevant tables.

The main verified characteristics of the components of metabolic syndrome: BMI, waist and hip circumference, baseline BP values, and lipid spectrum parameters, as well as clinically verified diagnoses of hypertension, obesity, DM2are supplemented and included in Table 3 and are presented in Table 1.

  1. I do not find the term "corrected", often used by authors, to be appropriate for all the questions and comments I have raised. Please respond to my points with sentences.

We apologize for the brevity. In this version, we have added explanations to your questions and comments.

  1. Does the difference in the proportion of people with diabetes between the two groups bias the results? Please explain your answer in detail.

We are grateful for the reviewer's attention and comments. We have once again reviewed the statistical analyses and identified an error in the data on the presence of diabetes mellitus in the patients (we attach a reference to our previous work with the same group of patients [37]). In addition, we would like to note that all patients with diabetes mellitus were initially (at baseline) compensated with pharmacological therapy.

  1. I maintain that the small sample size is a significant limitation of the present study. The authors plan to repeat their study with a larger sample size. What do you expect from analyses with larger sample sizes?

Yes, we agree that the main limiting factor is the number of patients. But this was a pilot study, which for the first time analyzed the microbiome, and in particular TMAO indicators, in the dynamics of hypoxic interventions. 

Using a larger sample size we would like to extend the study for more adequate randomization and to identify possible effects of different microbiome strains on TMAO values on the one hand, and on the other hand to link them with cardiovascular risk factors and other components of metabolic syndrome.

We plan to analyze in more detail the variations in microbiome strains and differences in their distribution in people with different levels of TMAO to confirm or refute the significance of TMAO as a prognostic indicator of cardiovascular event risk, and to examine changes with longer courses of hypoxic exposure training, since we suggest that changes are not instantaneous but take longer to develop. In addition, we would like to analyze the persistence of the effects in the long term.

In addition, we would like to profile the microbial communities of the gut microbiota of the experimental and control groups in terms of alpha and beta diversity and assess the stabilization of the microbiota composition after longer courses of hypoxic interventions. The differential representation of different microbial taxonomy before and after complex treatment with hypoxic training is currently being analyzed.

References:

  1. Bestavashvili, A. A.; Glazachev, O.S.; Bestavashvili, A.B.; Dhif, I.; Suvorov, A.Yu.; Vorontsov, N.V.; Tuter, D.S.; Gognieva, D.G.; Yong, Z.; Pavlov, C.S, Glushenkov, D.V.; Sirkina, E.A.; Kaloshina, I.V., and Kopylov P.Y. The Effects of Intermittent Hypoxic-Hyperoxic Exposures on Lipid Profile and Inflammation in Patients With Metabolic Syndrome. Front Cardiovasc Med. 2021;8:700826. Doi:10.3389/fcvm.2021.700826.

We thank the reviewer again for his careful consideration of our work!

Round 3

Reviewer 3 Report

Thanks to the authors for their answers to my questions/comments.

Author Response

Dear reviewer, we thank you for your attention and your comments to our work! Our team has made all the changes in accordance with your comments and notes. 

--

Afina Bestavashvili, MD.

I.M. Sechenov First Moscow State Medical University, Moscow, Russia